# Preparation and Strengthening Mechanisms of Ultrasonic-Assisted $Cr_3C_2$ Particle-Reinforced Al Matrix Composite

Yisong Xue [1], Wenyan Zhai [1,*], Xiao Li [1], Liang Sun [1,*], Yiran Wang [2], Yanming Liu [1] and Hui Dong [1]

1 College of Materials Science and Engineering, Xi'an Shiyou University, Xi'an 710065, China; xue1135780116@163.com (Y.X.); xli@xsyu.edu.cn (X.L.); ymliu@xsyu.edu.cn (Y.L.); donghui@xsyu.edu.cn (H.D.)
2 State Key Laboratory for Mechanical Behavior of Materials, Xi'an Jiaotong University, Xi'an 710049, China; wangyiran@xjtu.edu.cn
* Correspondence: 180606@xsyu.edu.cn (W.Z.); lsun@xsyu.edu.cn (L.S.); Tel./Fax: +86-029-8838-2598 (W.Z.)

**Abstract:** A novelty Al matrix composite reinforced with $Cr_3C_2$ particles was prepared by an ultrasound vibration treatment-assisted casting process. The $Cr_3C_2$ content on the microstructure and mechanical properties of $Cr_3C_{2p}$/Al composite were researched systematically. The results indicated that $Cr_3C_2$ particles were effectively distributed around the grain boundary and led to a great reduction in crystalline size. The density, porosity, and Vickers hardness of the composites rose simultaneously with the increase of $Cr_3C_2$ content. Meanwhile, the tensile strength and yield strength increased by 104.5% and 85.7%, respectively by adding 3.0 wt. % $Cr_3C_2$. The fracture morphologies showed that the fracture mechanism was plastic fracture for the pure Al and gradually transformed to brittle fracture as the content of $Cr_3C_2$ exceeded 3.0 wt. %. Moreover, the strengthening mechanism of the composite was also discussed.

**Keywords:** Al matrix composites; ultrasound; microstructure; mechanical properties



## 1. Introduction

Particle-reinforced metal matrix composites (PRMMCs) are widely used for structure applications because of their high specific strength and stiffness compared with conventional alloys, as well as their isotropic properties and relatively straightforward processing compared with continuous fiber-reinforced composites [1]. Particulate ceramic-reinforced Al matrix composite [2–5] possesses low density, high strength, and outstanding corrosion resistance, which has led to it being considered as a promising material and applied in the fields of electronic engineering, spacecraft, and petroleum engineering. As is generally known, the adhesive strength, wettability, and interfacial reaction of the Al matrix and reinforced particles have a significant effect on the mechanical properties of the composite. A stronger interface would usually allow for effective load transfer from the Al matrix to the reinforced particle, leading to improved strength, stiffness, and resistance to the external environment. Indeed, the pair of their thermal expansion coefficients should be matching between a couple of composites, which can effectively decrease the micro-cracks in the interface and enhance bonding strength during the fabrication process.

Potential reinforcement particles such as $Al_2O_3$ [6], $TiB_2$ [7], $Si_3N_4$ [8] and TiC [9] were commonly used to improve the microstructure and mechanical properties of Al matrix composite and three preparation methods, namely powder metallurgy [10], in-situ synthesis [11], and stirring casting [12], were available. Guo et al. [13] researched the influence of ceramic particle dimension on the mechanical performance of the $SiC_p$/Al matrix composite. The composite reinforced by 13 μm SiC ceramic particles (high hardness and insulator) possessed the highest hardness, while the highest strength was held by the

5 µm particle of the specimen. The hardness of the specimen accompanied by the strength declined due to the increase in ceramic particle size. Gao et al. [3] prepared the Al–Cu composite reinforced with $TiB_2$ ceramic using the in situ synthesis method and studied the effect of the ultrasonic stirring time on the microstructure, strength, and toughness of the composite. The experimental results indicated that ultrasonic stirring improved the possibility of subsequent reactions and formed more dispersive $TiB_2$. $TiB_2$ ceramic particle agglomeration was reduced and the crystalline size of the material declined with the increase of ultrasonic stirring time. Besides this, the yield strength was also improved by extending the ultrasonic stirring time. The microstructure and mechanical performance of the particulate-reinforced Al matrix composite fabricated by strongly ultrasonic stirring were superior to the one prepared by mechanical stirring. The Cu element's surface was coated with layer-structural $Ti_3AlC_2$ ceramic particles by Wang, which effectively improved the composite interface [14]. Then, the coated $Ti_3AlC_2$ was added into the Al matrix composite. The adhesive strength and mechanical properties were enhanced distinctly due to the generation of the $Al_2Cu$ transitional phase. Moreover, the tribological properties of the material were also improved, which was attributed to the increase in mechanical performance.

It can be seen that $TiB_2$ and $Ti_3AlC_2$ ceramics both exhibited metal-like electrical conductivity through fitting the interface with aluminum. Chromium carbides had three different crystallographic structures: orthorhombic $Cr_3C_2$, hexagonal $Cr_7C_3$, and cubic $Cr_{23}C_6$. Orthorhombic $Cr_3C_2$ possesses the best mechanical properties. $Cr_3C_2$ ceramic phase, a highly metallic carbide, was an outstanding potential material with high modulus, high hardness, high melting point, outstanding wear and corrosion resistance. $Cr_3C_2$-Ni composites had been fabricated in situ in our previous researche [15–17] and the physical performance, tribological properties, and oxidation resistance were studied systematically. The studied results demonstrated that strong interfacial bonding forces and good wettability were present in $Cr_3C_2$ and Ni. $Cr_3C_2$ can act as the heterogeneous nucleation of Ni. $Cr_3C_2$ particles had no interface reaction with aluminum. The coefficient of thermal expansion of $Cr_3C_2$ ceramic was similar to metal, like Al and Ni, which can effectually promote the interfacial bonding of ceramic and metallic materials.

Certainly, highly metallic $Cr_3C_2$ particles would be possible options to reinforce the Al matrix which may have an impressive combination of mechanical properties and good resistance to corrosion. Based on our previous research, $Cr_3C_2$ ceramic particles added into the Al matrix composite by mechanical stirring can effectively eliminate large agglomerations, but this is impossible for 50–100 µm small agglomerations, especially for small particles. Ultrasonic vibration is an outstanding technology to deal with molten-liquid aluminum and magnesium alloys [18,19]. Ultrasonic vibration treatment (UVT) can be used to improve the particulate distribution of composites by dispersing particles to create more effective reinforcements due to Orowan strengthening [20]. Unfortunately, few studies about using UVT to improve the particulate distribution of aluminum matrix composites reinforced with high volumes of $Cr_3C_2$ particles were available so far. The smaller $Cr_3C_2$ particles are easily agglomerated. In order to improve the situation, we used ultrasonic vibration treatment, which allowed the $Cr_3C_2$ particles to be more evenly distributed in the Al matrix.

In this work, a novelty Al matrix composite reinforced with $Cr_3C_2$ particles was prepared by an ultrasound vibration treatment-assisted casting process. Micro scale particles (1–5 µm) of $Cr_3C_2$ and the influence of its content on microstructure and mechanical properties of $Cr_2$ particle-reinforced Al matrix composite are highlighted in this paper. Besides this, the strengthening mechanism of the composite is also discussed in detail.

## 2. Experimental Procedure

### 2.1. Fabrication of $Cr_3C_2$ Particle-Reinforced Al Matrix Composite

In this paper, $Cr_3C_2$ particle-reinforced Al matrix composite was fabricated using pure Al ingot (99.99%) and $Cr_3C_2$ powder by an ultrasound-assisted casting process at 760 °C for

30 min. The details of the sketch of the UVT system can be found in reference material [3]. The vibration frequency of the UVT system was 20 ± 1 kHz. The purity and crystalline sizes of $Cr_3C_2$ powder were 99.9% and 1–5 μm, respectively. The holding time of ultrasonic vibration stirring was 10 min. A certain amount of $Cr_3C_2$ particle (1.0 wt. %, 2.0 wt. %, 3.0 wt. % and 4.0 wt. %) was added in the matrix. The molten liquid was poured into the graphite die after removing the contaminants on the surface of the molten liquid. The ingot casting was cut to the standard dimension for microstructure characterization and mechanical properties tests. The sample was ground and polished to remove traces of the linear cutting by using the waterproof abrasive paper from 800# to 7000#. The sample was then slightly etched for 1 min using the Kohler reagent to observe the microstructure of the composite.

### 2.2. Mechanical Properties Test

The density and porosity were tested by the Archimedean drainage method which had also been reported in reference material [21]. Vickers hardness (HV) tests were conducted using the hardness tester equipped with the diamond cone indenter, and the angle was 120°. The load and loading time were 60 kg and 10 s, respectively. The strength and toughness of the $Cr_3C_2$ particle-reinforced Al matrix composite were tested using the tensile testing machine INSTRON 1195 (INSTRON (Shanghai) Test Equipment Trading Co., Ltd., Shanghai, China). The tensile rate was 0.5 mm/min and the average value of three groups of tests was confirmed as the final value.

### 2.3. Materials Characterization

The detailed phases of the initial powder and the $Cr_3C_2$ particle-reinforced Al matrix composite were detected and demarcated by X-Ray Diffraction (XRD). Scanning electron microscopy (SEM) was applied to research the microstructure morphologies of the $Cr_3C_2$ particle-reinforced Al matrix composite. The distribution of different elements was detected by Energy Disperse Spectroscopy (EDS). The crystalline grain was dyed and the crystalline grain size was calculated by the software of Image-Pro plus 6.0.

## 3. Results and Discussion

### 3.1. Microstructure of $Cr_3C_{2p}$/Al Composite

The SEM figure and EDS result of the initial $Cr_3C_2$ powder can be seen the Figure 1a,b. The crystalline grain sizes of the $Cr_3C_2$ powder were 1–5 μm and the distribution of the powder is homogeneous. The EDS results indicate high purity of the $Cr_3C_2$ powder. Figure 1c shows the particle size distribution of the initial $Cr_3C_2$ powder. The percentages of the $Cr_3C_2$ powder with 0–1 μm, 1–2 μm, 2–3 μm, 3–4 μm, and 4–5 μm are 26%, 38%, 24%, 7%, and 5%, respectively. The average particle size of the initial $Cr_3C_2$ powder was 1.83 μm after calculation.

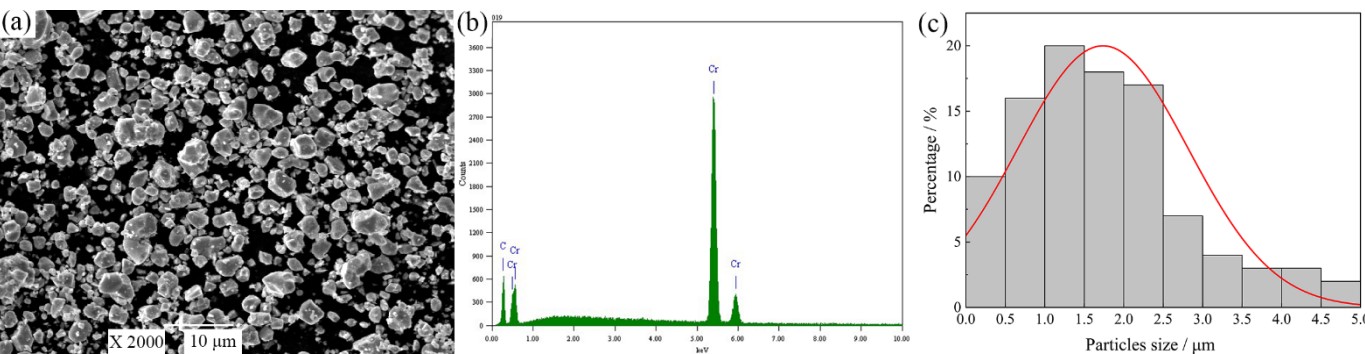

**Figure 1.** SEM figure, EDS result and particle size distribution of the initial $Cr_3C_2$ powder: (**a**) SEM figure, (**b**) EDS result, and (**c**) particle size distribution.

Figure 2a shows the XRD results of $Cr_3C_2$ particle-reinforced Al matrix composite. The enlarged drawing for the 4.0 wt. % $Cr_3C_{2p}$/Al sample is shown in Figure 2b. Compared with the pure Al sample, the diffraction peaks of $Cr_3C_2$ can be detected for the $Cr_3C_{2p}$/Al samples. Moreover, the diffraction peaks of Al shift to the left side, which can be attributed to the mutual diffusion of Cr and Al elements. The diffusion of Cr in Al will lead to an increase in the lattice constant and interplanar crystal spacing (d) of Al atoms. According to the Bragg law ($2d \cdot \sin\theta = n \cdot \lambda$), the diffraction angle ($\theta$) will decrease and the diffraction peaks of Al will shift to the left side.

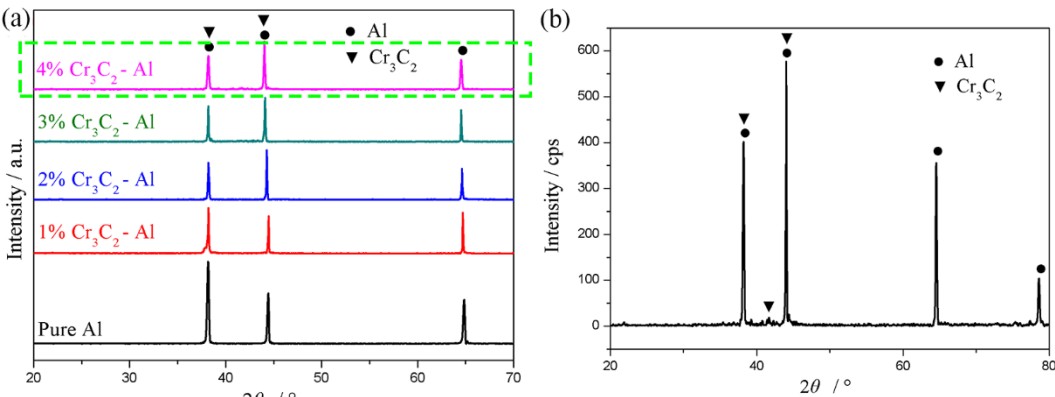

**Figure 2.** The XRD results of $Cr_3C_2$ particle-reinforced Al matrix composite: (**a**) XRD results and (**b**) the enlarged drawing for the 4.0 wt. % $Cr_3C_{2p}$/Al sample.

The SEM morphologies and dyed results of $Cr_3C_2$ particle-reinforced Al matrix composite are shown in Figure 3a–j. The average crystalline size of the pure Al sample is about 105 μm. The distribution of the microstructure becomes homogeneous with the increase of $Cr_3C_2$ content. Besides, the average crystalline size of the composite declines at the first stage and then increases slightly with the increase of $Cr_3C_2$ content. Some etch pits of $Cr_3C_2$ ceramic particles can be seen in the microstructure because of the slightly corrosion of Kohler reagent, as shown in Figure 3d,e. The grain shapes of the big $Cr_3C_2$ ceramic particles are granular, which well corresponds with the initial $Cr_3C_2$ powder in Figure 1. According to the computed results by using the software of Image-Pro plus 6.0, the average crystalline sizes of the 2.0 wt. % and 4.0 wt. % $Cr_3C_2$ particle-reinforced Al matrix composite samples are about 21 μm and 51 μm, respectively. Therefore, their average crystalline sizes decrease by 80% and 51.4%, respectively. The declined average crystalline grain size can improve simultaneously the strength and toughness of the $Cr_3C_2$ particle-reinforced Al matrix composite via grain refinement.

The element distributions of the 1.0 wt. % $Cr_3C_{2p}$/Al composite sample are also shown in Figure 4. The grain boundaries are clear and homogeneous. It can be observed that the Al element as the matrix can be detected distinctly (Figure 4b). As shown in Figure 4c,d, Cr and C elements congregate among the grain boundaries, indicating that $Cr_3C_2$ ceramic particles serve as the heterogeneous nucleus regions for the Al matrix during the process of solidification. The average crystalline grain size can be refined.

### 3.2. Mechanical Properties of $Cr_3C_{2p}$/Al Composite

Figure 5a–c represents the effects of $Cr_3C_2$ content on the density, porosity, and Vickers hardness of $Cr_3C_2$ particle-reinforced Al matrix composite, respectively. The densities of $Cr_3C_2$ and Al are 6.68 and 2.7 g/cm³, respectively. As we know, the hardness of $Cr_3C_2$ is higher than that of Al matrix. Logically, the density and hardness of the $Cr_3C_{2p}$/Al composite will rise simultaneously with the increase in $Cr_3C_2$ content. In contrast, the increase in porosity can be attributed to the increase in the grain boundaries. Figure 5d–f shows the SEM morphologies of the indenter for the pure Al, 2.0 wt. %, and 4.0 wt. % $Cr_3C_{2p}$/Al composite samples. The outlines of the indenter are clear and the micro-cracks

were not detected after the hardness tests, indicating excellent toughness of the $Cr_3C_{2p}/Al$ composite. The areas of the indenter decrease obviously with the increase in $Cr_3C_2$ content, which can also indicate the higher hardness of the $Cr_3C_2$ particle-reinforced Al matrix composite compared to pure Al. From Figure 5f, it can be seen that the $Cr_3C_2$ ceramic particles are removed because of the corrosion action of the Kohler reagent. Moreover, the micro-cracks are also not detected at the grain boundaries of $Cr_3C_2$ and Al, which can be attributed to the bonding strength interface.

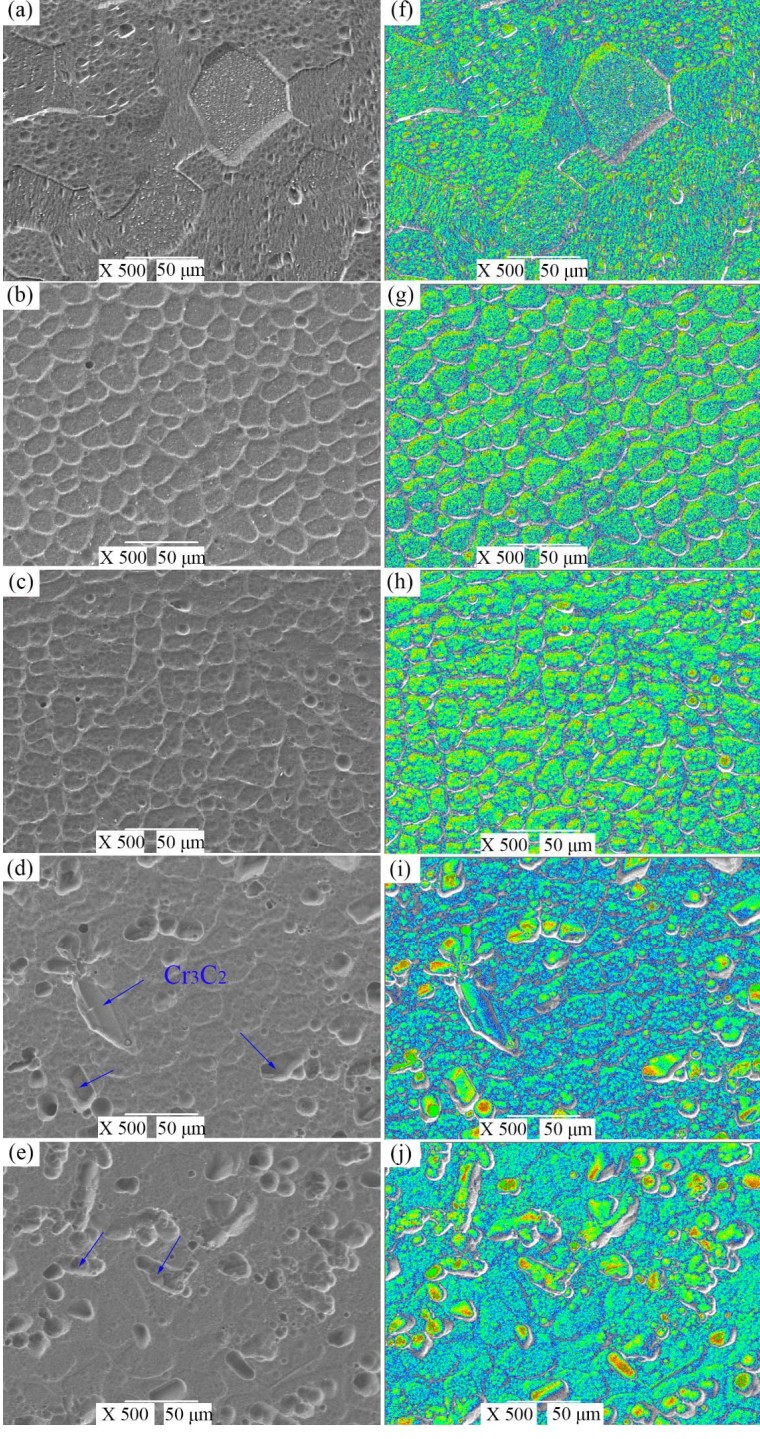

**Figure 3.** SEM morphologies and dyed results of $Cr_3C_2$ particle-reinforced Al matrix composite: (**a**,**f**) pure Al, (**b**,**g**) 1.0 wt. % $Cr_3C_{2p}/Al$ composite, (**c**,**h**) 2.0 wt. % $Cr_3C_{2p}/Al$ composite, (**d**,**i**) 3.0 wt. % $Cr_3C_{2p}/Al$ composite, and (**e**,**j**) 4.0 wt. % $Cr_3C_{2p}/Al$ composite.

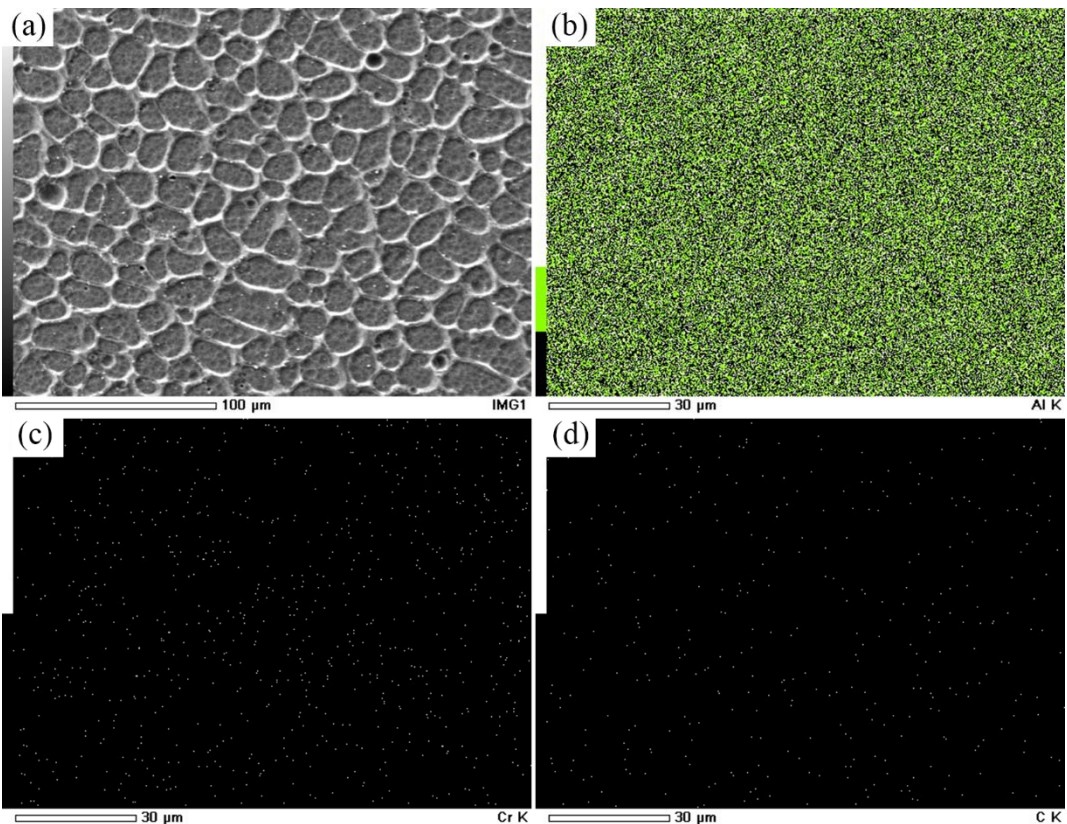

**Figure 4.** The element distributions of the 1.0 wt. % $Cr_3C_{2p}$/Al composite sample: (**a**) microstructure, (**b**) Al element, (**c**) Cr element, and (**d**) C element.

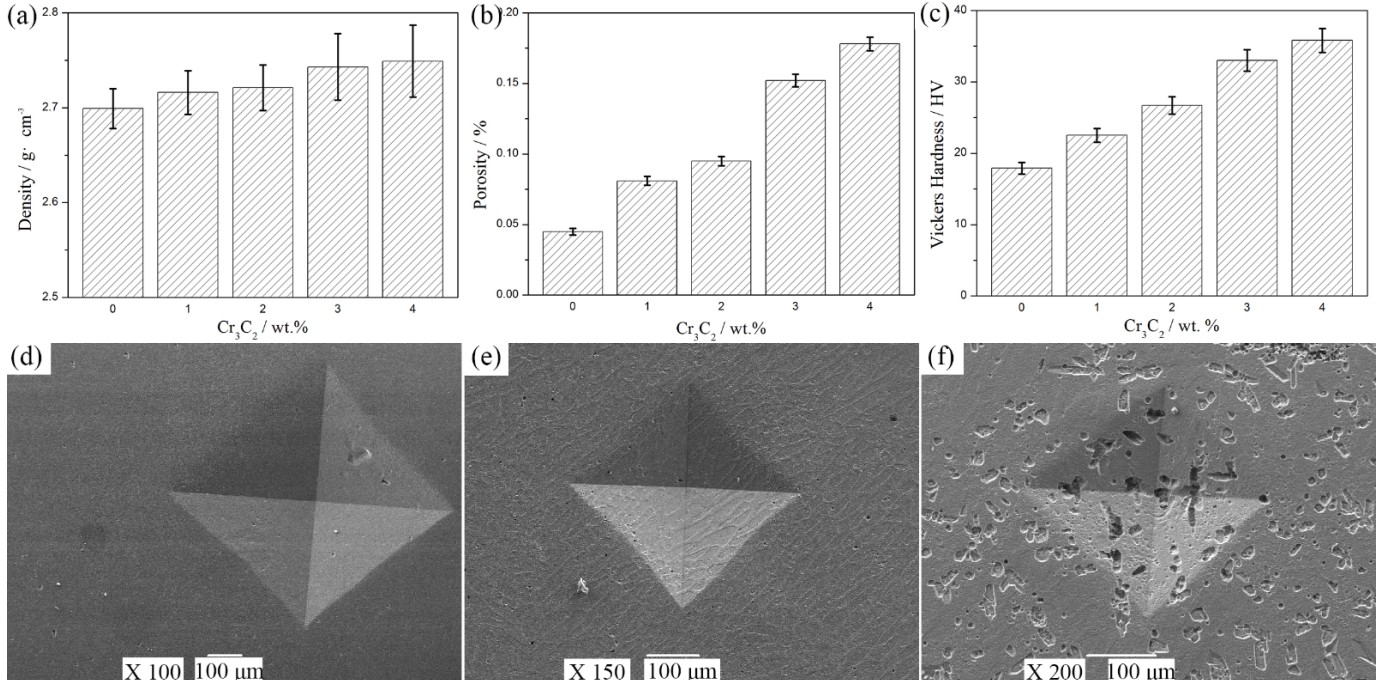

**Figure 5.** The mechanical properties and the indenter morphologies of the $Cr_3C_2$ particle-reinforced Al matrix composite: (**a**) density, (**b**) porosity, (**c**) Vickers hardness, (**d**) pure Al, (**e**) 2.0 wt. % $Cr_3C_{2p}$/Al composite, and (**f**) 4.0 wt. % $Cr_3C_{2p}$/Al composite.

Figure 6a shows the stress–strain curves of the $Cr_3C_2$ particle-reinforced Al matrix composite with different $Cr_3C_2$ amounts added. Obviously, the stress rises, while the strain

declines with the addition of $Cr_3C_2$ ceramic particles. Pure Al possesses the best toughness, and the toughness declines rapidly as the $Cr_3C_2$ content exceeds 3.0 wt. %. The 3.0 wt. % $Cr_3C_{2p}$/Al composite sample behaves with the best tensile strength and yield strength. As shown in Figure 6b,c, the tensile strengths of pure Al, 1.0 wt. %, 2.0 wt. %, 3.0 wt. %, and 4.0 wt. % $Cr_3C_{2p}$/Al composite samples are 44 MPa, 62 MPa, 73 MPa, 90 MPa, and 79 MPa, respectively. Their yield strengths are 26.3 MPa, 33.1 MPa, 37.4 MPa, 48.8 MPa, and 45.7 MPa, respectively. Compared with the pure Al sample, the tensile strengths of 1.0 wt. %, 2.0 wt. %, 3.0 wt. %, and 4.0 wt. % $Cr_3C_{2p}$/Al composite samples increase by 40.9%, 65.9%, 104.5%, and 79.5%, respectively. Therefore, it can be concluded that the strength is enhanced distinctly due to the addition of the $Cr_3C_2$ ceramic particles.

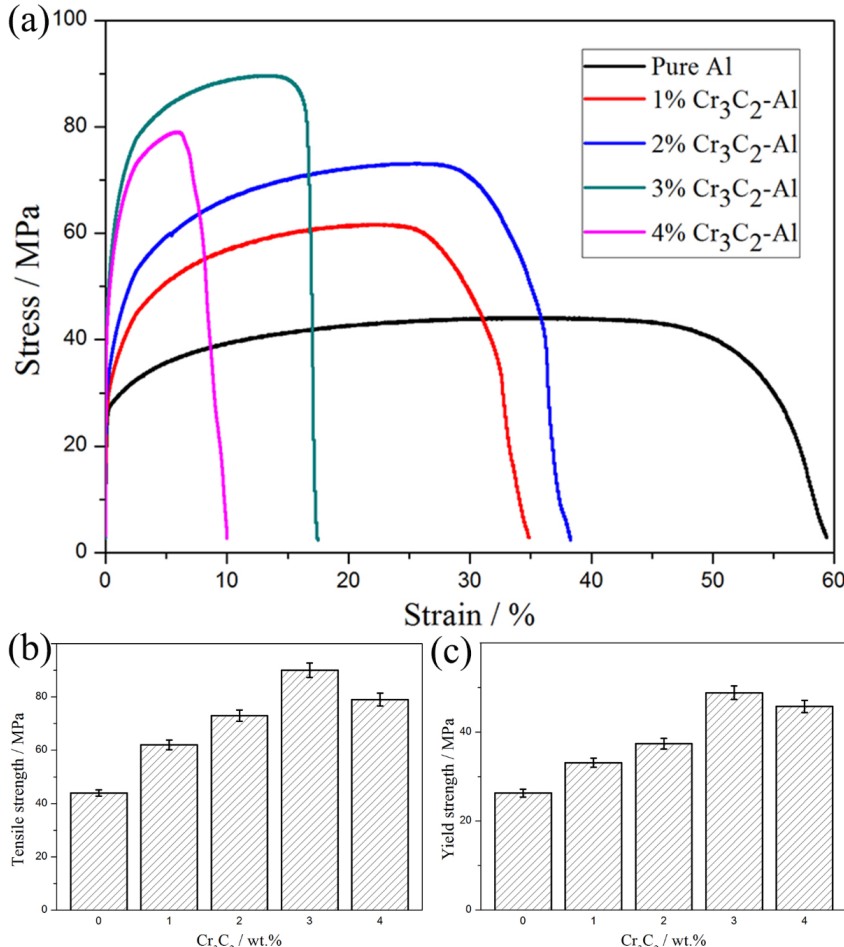

**Figure 6.** The stress-strain curves, tensile strength, and yield strength of the $Cr_3C_2$ particle-reinforced Al matrix composite with different $Cr_3C_2$ amounts added: (**a**) stress–strain curves, (**b**) tensile strength and (**c**) yield strength.

### 3.3. Fracture Morphology

The fracture morphologies of the $Cr_3C_2$ particle-reinforced Al matrix composite samples are shown in Figure 7a–j. Figure 7a shows the fracture morphologies of the pure Al sample. There is no impurity element for the pure Al sample and it can be seen that the fracture is smooth. From Figure 7a–e, it can be observed that the area of the fracture morphology rises distinctly with an increase of the $Cr_3C_2$ content, which is in good agreement with the result of the section shrinkage rate. $Cr_3C_2$ ceramic particles and large dimples can be seen in Figure 7g,h, indicating that the 1.0 wt. % and 2.0 wt. % $Cr_3C_{2p}$/Al composite samples exhibit excellent ductility. However, the sizes of the dimples decrease as the $Cr_3C_2$ content exceeds 3.0 wt. %. Moreover, the micro-cracks and cleavage steps exist on the fracture morphologies for the 3.0 wt. % and 4.0 wt. % $Cr_3C_{2p}$/Al composite samples,

as labeled in Figure 7i,j. The generation and extension of the micro-cracks will result in fracture mechanism changes from ductile fracture to brittle fracture.

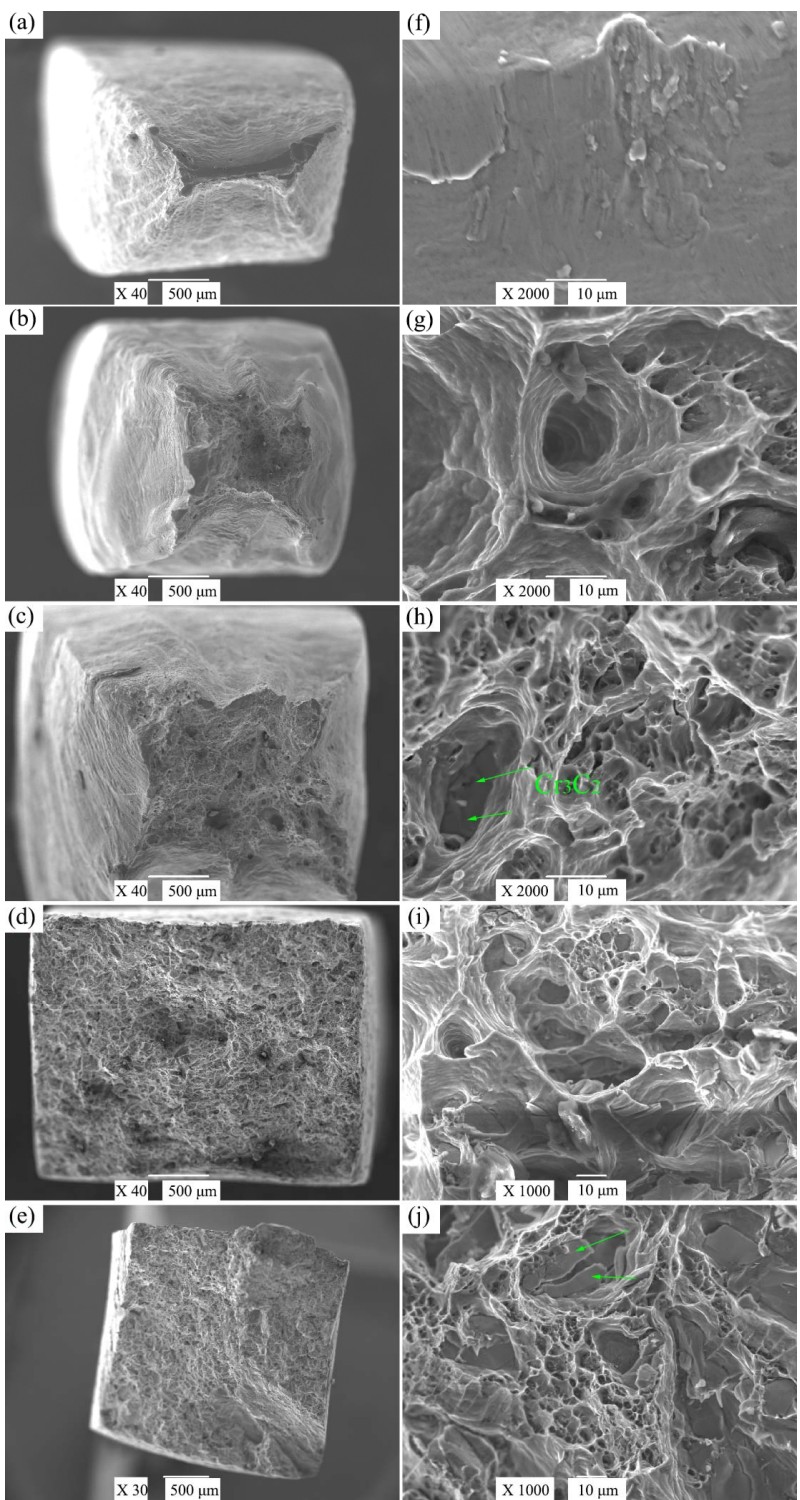

**Figure 7.** The fracture morphologies of the $Cr_3C_2$ particle-reinforced Al matrix composite: (**a**,**f**) pure Al, (**b**,**g**) 1.0 wt. % $Cr_3C_{2p}$/Al composite, (**c**,**h**) 2.0 wt. % $Cr_3C_{2p}$/Al composite, (**d**,**i**) 3.0 wt. % $Cr_3C_{2p}$/Al composite, and (**e**,**j**) 4.0 wt. % $Cr_3C_{2p}$/Al composite.

Figure 8 represents the effects of the $Cr_3C_2$ content on the elongation and section shrinkage rate of the $Cr_3C_2$ particle-reinforced Al matrix composite. The tensile strength

and yield strength increase distinctly as mentioned above, while the elongation and section shrinkage rate decline with the addition of $Cr_3C_2$ ceramic particles. The 2.0 wt. % $Cr_3C_{2p}$/Al composite sample exhibits the best elongation and the 1.0 wt. % $Cr_3C_{2p}$/Al composite sample shows the best section shrinkage rate by adding $Cr_3C_2$ ceramic particles. From Figure 8a,b, it can be observed that the elongation and section shrinkage rate decline obviously when the addition of $Cr_3C_2$ ceramic particle exceeding 3.0 wt. %. The declining ductility will transform the fracture mechanism from ductile fracture to brittle fracture.

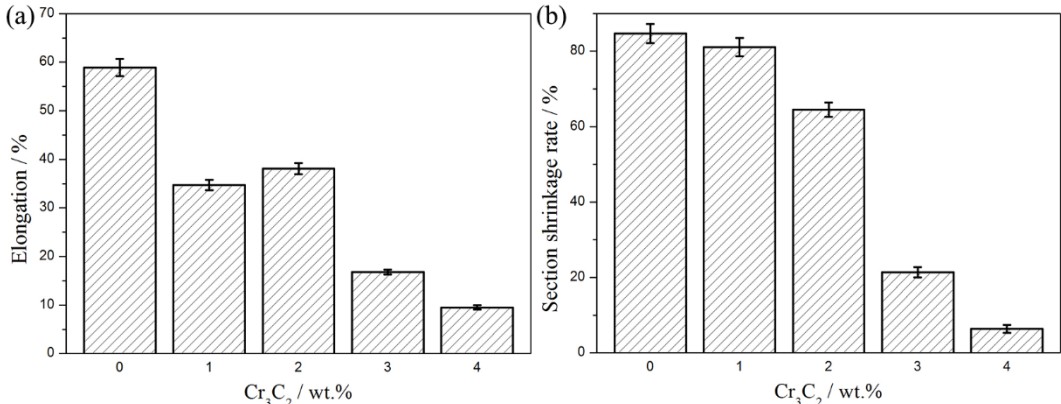

**Figure 8.** The elongation and section shrinkage rate of the $Cr_3C_2$ particle-reinforced Al matrix composite: (**a**) elongation and (**b**) section shrinkage rate.

### 3.4. Strengthening Mechanism

There are four accessible strengthening mechanisms [22–26] for the enhancement of the yield strength of the $Cr_3C_2$ particle-reinforced Al matrix composite: grain refinement strengthening ($\sigma_g$), load-bearing strengthening ($\sigma_{Load}$), CTE mismatch strengthening ($\sigma_{CTE}$), and Orowan strengthening ($\sigma_{Orowan}$). Orowan strengthening is mainly because of the dislocation movement, which can go through or around the second phase particles. However, the limit of Orowan strengthening [27] is the crystalline grain sizes of the second phase particles, which should be less than 1 μm. In this study, the crystalline grain sizes of $Cr_3C_2$ ceramic particles are 1–5 μm. Therefore, Orowan strengthening is not appropriate for this study. The final predicted result of yield strength ($\sigma_p$) can be calculated by the following equation:

$$\sigma_p = \sigma_m + \sigma_g + \sigma_{Load} + \sigma_{CTE} \tag{1}$$

where $\sigma_m$ is the yield strength of the Al matrix. In this study, $\sigma_m$ is 26.3 MPa. As we know, grain refinement strengthening ($\sigma_g$) can be calculated by the Hall–Petch equation [28] as following:

$$\sigma_g = k(d_c^{-1/2} - d_m^{-1/2}) \tag{2}$$

where k is the strengthening factor, $d_c$ is the average crystalline grain size of the $Cr_3C_2$ particle-reinforced Al matrix composite, and $d_m$ is the average crystalline grain size of the Al matrix. According to the reference [29], the strengthening factor of Al is 68 MPa $\mu m^{1/2}$. In this study, the average crystalline grain sizes of pure Al, 1.0 wt. %, 2.0 wt. %, 3.0 wt. %, and 4.0 wt. % $Cr_3C_{2p}$/Al composite samples are 105 μm, 23 μm, 21 μm, 34 μm, and 51 μm, respectively.

During the experimental process, the load has an important factor on the yield strength of the $Cr_3C_2$ particle-reinforced Al matrix composite. The load-bearing strengthening ($\sigma_{Load}$) can be calculated by the following equation [30]:

$$\sigma_{Load} = 1/2V_p\sigma_m \tag{3}$$

where $V_p$ is volume fraction of $Cr_3C_2$ ceramic particles. In this study, the volume fractions of 1.0 wt. %, 2.0 wt. %, 3.0 wt. %, and 4.0 wt. % $Cr_3C_{2p}/Al$ composite samples are 0.41%, 0.82%, 1.23%, and 1.66%, respectively.

The difference of the coefficient of thermal expansion (CTE) between $Cr_3C_2$ ceramic particles and the pure Al matrix will easily generate dislocations during the solidification process, which also has a significant effect on the yield strength of the $Cr_3C_2$ particle-reinforced Al matrix composite. In this study, the coefficient of thermal expansion of $Cr_3C_2$ ceramic particle and pure Al are $10.3 \times 10^{-6}$ K$^{-1}$ and $23.6 \times 10^{-6}$ K$^{-1}$, respectively. The CTE mismatch strengthening ($\sigma_{CTE}$) can be calculated by the following equations [31]:

$$\sigma_{CTE} = \beta G_m b(\rho_{CTE})^{1/2} \tag{4}$$

$$\rho_{CTE} = 12\Delta\alpha\Delta TV_p/(bd_p(1 - V_p)) \tag{5}$$

where $\beta$ is the constant value of 1.25, and $G_m$ is the shear modulus of pure Al; it can be calculated by $G_m = E_m/(2(1 + v))$, in which $E_m$ and $v$ are the Young modulus and Poisson's ratio of pure Al, respectively. $\Delta\alpha$ is the difference value of the coefficient of thermal expansion of the pure Al and $Cr_3C_2$ ceramic particles, $\Delta T$ is the difference value of the casting temperature and room temperature, $V_p$ is the volume fraction of $Cr_3C_2$ ceramic particle, b is the Burgers vector with the value of 0.286 nm, and $d_p$ is the average crystalline grain size of the $Cr_3C_2$ particles. The detailed parameter values of them for the enhancement of the yield strength of $Cr_3C_2$ particle-reinforced Al matrix composite are shown in Table 1.

**Table 1.** The detailed parameter values for the CTE mismatch strengthening of the $Cr_3C_{2p}/Al$ composite.

| Sample | $d_p$/nm | $V_p$ | $\beta$ | $G_m$/GPa | $E_m$/GPa | $v$ | b/nm | $\Delta\alpha/10^{-6}$ K$^{-1}$ | $\Delta$T/K |
|---|---|---|---|---|---|---|---|---|---|
| 1.0 wt. % $Cr_3C_{2p}/Al$ | | 0.41% | | | | | | | |
| 2.0 wt. % $Cr_3C_{2p}/Al$ | 3200 | 0.82% | 1.25 | 26.3 | 70 | 0.33 | 0.286 | 13.3 | 735 |
| 3.0 wt. % $Cr_3C_{2p}/Al$ | | 1.23% | | | | | | | |
| 4.0 wt. % $Cr_3C_{2p}/Al$ | | 1.66% | | | | | | | |

The final predicted result and experimental result of the yield strength of $Cr_3C_{2p}/Al$ composite are shown in Table 2. From this table, it can be seen that the predicted result of the yield strength is in good accordance with the experimental result for the $Cr_3C_{2p}/Al$ composite samples. Moreover, it can be concluded that the main strengthening mechanisms for the improvement of the yield strength are grain refinement strengthening ($\sigma_g$) and CTE mismatch strengthening ($\sigma_{CTE}$).

**Table 2.** The final predicted result and experimental result of the yield strength.

| Sample | $\sigma_m$/MPa | $\sigma_g$/MPa | $\sigma_{Load}$/MPa | $\sigma_{CTE}$/MPa | $\sigma_p$/MPa | Measured YS/MPa |
|---|---|---|---|---|---|---|
| 1.0 wt. % $Cr_3C_{2p}/Al$ | 26.3 | 7.5 | 0.054 | 6.8 | 40.6 | 33.1 |
| 2.0 wt. % $Cr_3C_{2p}/Al$ | 26.3 | 8.2 | 0.108 | 9.7 | 44.3 | 37.4 |
| 3.0 wt. % $Cr_3C_{2p}/Al$ | 26.3 | 5.0 | 0.162 | 11.9 | 43.4 | 48.8 |
| 4.0 wt. % $Cr_3C_{2p}/Al$ | 26.3 | 2.9 | 0.218 | 13.8 | 43.2 | 45.7 |

## 4. Conclusions

In this paper, a novel Al matrix composite reinforced with $Cr_3C_2$ particles was prepared by an ultrasound-assisted casting process at 760 °C for 30 min. The effects of $Cr_3C_2$ content on the microstructure and mechanical properties of $Cr_3C_{2p}/Al$ composite were researched systematically for the first time. The conclusions could be obtained as following:

1.  Compared with the pure Al sample, the average crystalline sizes of 1.0 wt. % $Cr_3C_{2p}/Al$ composite decreased by 80%. The decreased average crystalline grain size can simul-

taneously improve the strength and toughness of the $Cr_3C_{2p}$/Al composite via grain refinement strengthening.

2.  The fracture morphologies showed that the fracture mechanism was plastic fracture for the pure Al, 1.0 wt. %, and 2.0 wt. % $Cr_3C_{2p}$/Al composite samples, which gradually transformed to brittle fracture as the content of $Cr_3C_2$ exceeded 3.0 wt. %.

3.  After calculation, the predicted result of the yield strength was in good accordance with the experimental result for the $Cr_3C_{2p}$/Al composite samples. The main strengthening mechanisms for the improvement of the yield strength were grain refinement strengthening ($\sigma_g$) and CTE mismatch strengthening ($\sigma_{CTE}$).

**Author Contributions:** Conceptualization, W.Z. and L.S.; methodology, W.Z.; software, X.L.; validation, Y.X. and Y.W.; formal analysis, Y.L.; investigation, H.D.; resources, W.Z.; data curation, Y.X.; writing—original draft preparation, W.Z.; writing—review and editing, W.Z.; visualization, L.S.; supervision, X.L.; project administration, Y.L.; funding acquisition, W.Z. All authors have read and agreed to the published version of the manuscript.

**Funding:** This research was funded by [W. Zhai] grant number [20192110, HKDNM2019018, 2020JQ-777 and 20JK0837], [L. Sun] grant number [20202212, HKDNM201811 and 2019JQ-821] and [Y. Xue] grant number [YCS21111016].

**Institutional Review Board Statement:** The study was approved by the Institutional Review Board of Xi'an Shiyou University.

**Informed Consent Statement:** Not applicable.

**Data Availability Statement:** Not applicable.

**Acknowledgments:** This work was supported by the Open Fund of State Key Laboratory for Mechanical Behavior of Materials (20192110 and 20202212), the Open Fund of National Joint Engineering Research Center for Abrasion Control and Molding of Metal Materials (HKDNM201811 and HKDNM2019018), the Natural Science Basic Research Plan in Shaanxi Province of China (2019JQ-821 and 2020JQ-777), the Scientific Research Program Funded by Shaanxi Provincial Education Department (20JK0837) and the Graduate Student Innovation and Practical Ability Training Program of Xi'an Shiyou University (YCS21111016).

**Conflicts of Interest:** The authors declare no conflict of interest.

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
