# Peer review of "Preparation and Strengthening Mechanisms of Ultrasonic-Assisted Cr3C2 Particle-Reinforced Al Matrix Composite"

_coatings, doi:10.3390/coatings12040459_

Round 1

Reviewer 1 Report

1) Kindly please enhance the language standard

2) In CR3C2, 3 and 2 should be subscripted

3) Kindly please include some references published last 3 years related to Strengthening Mechanisms and effect of reinforcement particles in aluminium composites. You may use the following.

X Na, L Wenqing, et al “Effect of Scandium in Al-Sc and Al-Sc-Zr Alloys under precipitation strengthening mechanism at 3500C Aging,” Metals and Materials International, 27(12), 5145–5153, 2021.

Cao Fenghong, Chen Chang, Wang Zhenyu, et al “Effects of Silicon Carbide and Tungsten Carbide in Aluminium Metal Matrix Composites,” Silicon, 11(6), 2625–2632, 2019.

 Shoufa Liu, Yinwei Wang, et al “Effect of B4C and MOS2 reinforcement on micro structure and wear properties of aluminum hybrid composite for automotive applications,” Composites Part B: Engineering, 176, 107329, 2019.

M Manoj, GR Jinu, et al,“Multi response optimization of AWJM process parameters on machining TiB2 particles reinforced Al7075 composite using Taguchi-DEAR methodology,” Silicon, 10 (5), 2287-2293, 2018.

4) The novelty can clearly mentioned as last paragraph in introduction section

5) The methodology is neatly explained.

6) What vibration frequency range?

7) What is the load applied over specimen during vickers hardness testing

8) The conclusion section may be refined.

Reviewer 2 Report

Comments to authors sent to Editor separately.

Round 2

Reviewer 2 Report

The paper is improved after revision. Nevertheless, some results need additional comments:

  1. Fig.2,a is totally new one. The additional comment is added into the text which is attempting to explain the peek shift by solid solution formation. But there is no shift in the new picture visible. If authors insist they should add a table with interatomic   distances in order to confirm the fact of solid solution formation.
  2. Fig.4 contains pictures c) and d) which are totally black, at least in my copy.
  3. Fig.8 and comments compare elongation, i.e. maximum elongation at the end of deformation experiment, with shrinkage rate (!). Any rate include time for measuring its value. In Fig.8,b the sectional rate for the sample shrinkage is given in %. What does it mean? Probably, it would be better to characterize the deformation process by maximum shrinkage, not rate? Elongation and shrinkage are linked each other through the volume of a sample which is remain constant until the sample is distroyed (at maximum elongation). In the present form Fig.8,a do not coincide with the monotonic data from Fig.8,b.